# Investigation of the Temperature Compensation of Piezoelectric Weigh-In-Motion Sensors Using a Machine Learning Approach

**DOI:** 10.3390/s22062396

**Published:** 2022-03-20

**Authors:** Hailu Yang, Yue Yang, Yue Hou, Yue Liu, Pengfei Liu, Linbing Wang, Yuedong Ma

**Affiliations:** 1National Center for Materials Service Safety, University of Science and Technology Beijing, Beijing 100083, China; g20199181@xs.ustb.edu.cn; 2College of Metropolitan Transportation, Beijing University of Technology, Beijing 100124, China; yuehou@bjut.edu.cn; 3The Key Laboratory of Urban Security and Disaster Engineering of Ministry of Education, Beijing University of Technology, Beijing 100124, China; yliu@bjut.edu.cn; 4Institute of Highway Engineering (ISAC), RWTH Aachen University, D52074 Aachen, Germany; liu@isac.rwth-aachen.de; 5Joint USTB Virginia Tech Lab on Multifunctional Materials, USTB, Virginia Tech, Department Civil & Environmental Engineering, Blacksburg, VA 24061, USA; 6Beijing Municipal Construction Engineering Co., Ltd., Beijing 100141, China; myd780521@163.com

**Keywords:** piezoelectric sensor, temperature compensation, GA-BP neural network, Weigh-In-Motion, error analysis

## Abstract

Piezoelectric ceramics have good electromechanical coupling characteristics and a high sensitivity to load. One typical engineering application of piezoelectric ceramic is its use as a signal source for Weigh-In-Motion (WIM) systems in road traffic monitoring. However, piezoelectric ceramics are also sensitive to temperature, which affects their measurement accuracy. In this study, a new piezoelectric ceramic WIM sensor was developed. The output signals of sensors under different loads and temperatures were obtained. The results were corrected using polynomial regression and a Genetic Algorithm Back Propagation (GA-BP) neural network algorithm, respectively. The results show that the GA-BP neural network algorithm had a better effect on sensor temperature compensation. Before and after GA-BP compensation, the maximum relative error decreased from about 30% to less than 4%. The sensitivity coefficient of the sensor reduced from 1.0192 × 10^−2^/°C to 1.896 × 10^−4^/°C. The results show that the GA-BP algorithm greatly reduced the influence of temperature on the piezoelectric ceramic sensor and improved its temperature stability and accuracy, which helped improve the efficiency of clean-energy harvesting and conversion.

## 1. Introduction

In order to control vehicle overload, monitor and evaluate traffic volume, and ensure the safety and efficiency of vehicle transportation, the highway Weigh-In-Motion (WIM) system was produced [1,2]. The WIM system automatically obtains and detects the vehicle load information by embedding load cells on the driving road without affecting the normal driving of the vehicle, so as to fulfill its purpose of overload control [3,4]. The core component of the vehicle WIM system is the weighing sensor. At present, the products on the market have low measurement speed and low detection accuracy, with an average error of ±5–±30% [5,6,7]. While the American Society for testing and materials’ (ASTM) specification standard (ASTME1318) stipulates that the total weight error of the high-speed weighing system (Class III) is 6%, and the total weight error of the European WIM system is uniformly specified as 5%, China’s current road dynamic weighing standards are generally consistent with ASTM and European standards, and the total weight error of vehicles in the high- and low-speed-control weighing system is 5% [8]. It can be seen that the accuracy of WIM sensors needs to be improved in a large range. In addition, vehicle monitoring of some key expressways, bridges and culverts in China generally introduces foreign products [6], which cannot be widely used in China’s highway network because of their high price. There is a variety of WIM equipment on the market. Piezoelectric ceramic sensors have gradually shown their potential in WIM systems [9] because of their good dynamic characteristics, simpler installation process, small volume, low cost, high reliability and good structural compatibility [10,11,12]. When a piezoelectric ceramic sensor is used as WIM equipment to monitor vehicle overload, ensuring the measurement accuracy of the vehicle WIM system, improving the dynamic measurement range and reducing costs are the main problems it faces [13]. However, piezoelectric ceramic itself has temperature-sensitive characteristics, and the sensor, made of piezoelectric ceramic material, has temperature drift [14,15,16]. This drift will make the performance of the sensor unstable in the environment’s changing temperature, and then affect the detection accuracy [17]. Therefore, research on an effective temperature compensation method is of great significance to improve the detection accuracy of piezoelectric ceramic sensors. Haider et al. [18] studied the impact of a WIM system monitoring error changes on rigid and flexible pavement, and concluded that a positive deviation of 10% will lead to an overestimation of pavement life of about 5%; contrarily, a negative deviation of 10% will lead to an overestimation of pavement life of about 30% to 40%. Prozi and Hong [19] reached similar conclusions in their work, which further confirmed the importance of WIM data accuracy. Burnos and Gajda [20] analyzed, in detail, the thermal characteristics of sensors installed on the road, and also confirmed that temperature is an important factor affecting the accuracy of weighing results.

In order to compensate for the influence of temperature on piezoelectric sensors and improve their detection accuracy and sensitivity, we carried out temperature-compensation experimental research on a self-designed piezoelectric ceramic sensor. The data were processed by establishing a mathematical model of the temperature characteristics of the piezoelectric ceramic sensor, and using a Genetic Algorithm Back Propagation (GA-BP) neural network model. The experimental results show that the GA-BP neural network temperature compensation model can better improve the measurement accuracy and sensitivity of the piezoelectric ceramic sensor.

In practical engineering, temperature drift has become the biggest problem in the practical application of pressure sensors [9]. Research on temperature compensation has always been a hot topic for civil engineers [21,22,23,24]. Software compensation is often used in temperature compensation algorithms [25], and has the advantages of high precision, low cost and good universality. Software compensation methods can be divided into two categories: one is a numerical analysis method, and the other is a machine learning method [26]. For the former, the commonly used temperature compensation algorithms include the least square method, multi terminal broken line approximation method and multiple regression analysis method. For the latter, neural network is the most commonly used temperature method based on artificial intelligence [27]. Neural network has been widely considered and studied in the field of temperature compensation because of its strong generalization ability, good fault tolerance and strong nonlinear mapping ability. In this paper, a polynomial regression algorithm and BP neural network are proposed to realize the temperature compensation of the pressure sensor; however, the BP neural network has some shortcomings, such as a slow learning rate and susceptibility to falling into the local minimum [28,29]. Therefore, a genetic algorithm (GA) can be used to optimize the shortcomings of the BP neural network before the BP network training parameters [30,31,32,33]. Zhang et al. [22] used the improved BP neural network model to correct the temperature of an EME sensor. The results showed that the maximum relative error of internal force measurement was within 0.9% in the range of 10 °C to 60 °C. Wang et al. [23] corrected the temperature of a MEMS sensor in the temperature range of −10 to 80 °C using a neural network optimized by a genetic algorithm. The results showed that the maximum error of this method was 0.017%.

## 2. Design and Fabrication of Piezoelectric Ceramic Sensor

Based on previous research [34,35,36,37], this paper developed and designed a new piezoelectric ceramic sensor with the advantages of excellent environmental adaptability, high sensitivity and low cost. It can be used as Weigh-In-Motion sensor for road traffic monitoring. The piezoelectric material used in the sensor was lead zirconate titanate (PZT). The model of piezoelectric ceramics was PZT-4 (Hongsheng Acoustic Electronic Equipment Co., Ltd., Baoding, China). The packaging material was glass fiber nylon (PA66+30% GF) (Xingdeli Technology Co., Ltd., Shenzhen, China). The two-component epoxy resin was Kafuter K-9741(Xiaoka e-commerce Co., Ltd., Changzhou, China). The parameters of the material were provided by the supplier and are shown in Table 1, Table 2 and Table 3.

As shown in Figure 1, the piezoelectric sensor packaging slot adopted a band-type encapsulation structure. The overall size of the sensor was 150 mm × 45 mm × 28 mm (L × W × H). There was a reserved groove with a diameter φ of 25 mm for the PZT patch position in the middle of the protective structure. The size of the PZT patch was φ 20 mm × 2 mm. A stainless-steel gasket was added between the piezoelectric ceramic and the groove bottom in order to make the piezoelectric ceramic stress evenly. The size of the stainless-steel gasket was φ 25 mm × 2 mm.

Based on the above design, the manufacturing process of the sensor was as follows: Firstly, we placed the stainless-steel gasket in the reserved groove and bonded the piezoelectric ceramic sheet to the stainless-steel gasket with quick-drying glue. Secondly, the signal wires were connected with the positive and negative pole of the piezo sheet and fixed into the groove wall. The remaining wires were drawn from the groove. Thirdly, the flexible electronic silica gel was poured into the gap between the groove wall and the piezoelectric ceramic, which could effectively prevent the piezoelectric material and electrode from being affected by water or steam. Finally, after the flexible electronic gel solidified, the two-component epoxy resin was filled and the air tightness and flatness were met. Using the same production process, two sets of sensors were packaged as SP-1 and SP-2, as shown in Figure 2.

## 3. Temperature Characteristics Study of Piezoelectric Ceramics

### 3.1. Temperature Characteristics of Piezoelectric Ceramics

PZT-4 has the advantage of a high piezoelectric coefficient, and is an ideal force-sensing material. The performance parameters of piezoelectric ceramics are also different at different temperatures, which will result in changes in the piezoelectric sensor output and affect the accuracy of the sensor. This section tested the performance parameters of piezoelectric ceramics at different temperatures. Through the mechanical test, the sensor output signals under different temperature loads were obtained as the data basis for subsequent temperature correction.

First of all, the variation in electrical parameters with temperature were measured. Five PZT-4 piezoelectric ceramic sheets of the same size (φ 20 mm × 1.5 mm) were randomly selected. Temperature-adjustable refrigerators provided sub-zero temperatures with a temperature range of −40–0 °C, and the vacuum drying oven provided above zero temperatures in the range of 0–250 °C. In this test, the temperature was controlled at −25–80 °C, at an interval of 15 °C. The temperature was kept for 1h after the equipment reached the set temperature. Then, the specimens were taken out and measured in 1 min. The average values of the measured data of five samples were taken. Then, the data were used to analyze the changes in the piezoelectric coefficient and the capacitance of the piezoelectric ceramics with the change in temperature. The piezoelectric constant d_33_ was measured using a quasi-static piezoelectric coefficient measuring instrument ZJ-6A, and the capacitance was measured using a UT603 capacitance inductance meter. The test equipment is shown in Figure 3, and the measurement results are shown Figure 4.

From Figure 4, it can be seen that the environmental temperature has a great influence on the electrical properties of PZT. In the temperature range of −25 °C to 80 °C, the piezoelectric coefficient and the capacity of the PZT-4 increased with the temperature, but the relationship between them is obviously not linear.

### 3.2. Test Design of Temperature Compensation

There is a linear relationship between the ideal pressure sensor output charge and the measured pressure when the temperature is constant. However, in practical application, there is a nonlinear relationship between the pressure sensor output u and the input force F. Therefore, within the working temperature range of the pressure sensor, different temperatures can be selected to calibrate the compensated pressure sensor. In order to better test the influence of experimental temperature on the PZT sensor, the experimental control temperature was −25 °C to 80 °C, which covers most road conditions. The test temperature was successively set to −25 °C, −10 °C, 5 °C, 20 °C, 35 °C, 50 °C, 65 °C and 80 °C. According to the standard axle load and the load range of the machine specified in the code for the design of the highway asphalt pavement (JTG D50-2017), the loading force range was 5–25 kN. The load increment interval was 2 kN. At each temperature level, the temperature lasted for one hour to ensure a uniform temperature inside the sensor, and then the data were recorded. The test loading equipment adopted the electro-hydraulic servo fatigue testing machine from the Walter+Bai test equipment company in Switzerland, with a load range of ±100 kN and a temperature-control-box range of −150 °C to 350 °C. The loading test scheme is shown in Figure 5a. The loading device and the data acquisition equipment are shown in Figure 5 b and c. The load was sinusoidal, with amplitudes ranging from 5 to 25 kN, corresponding to an vehicle axle load range of 20–125 kN. The loading frequency was 5 Hz, which corresponded to a vehicle speed of 20–30 km/h. The output signal of the sensors was charge, which was converted into a voltage signal through a charge amplifier and amplified to an appropriate multiple. Finally, the data were collected by oscilloscope.

The digital oscilloscope type was a DPO2024 from Tektronix and the charge amplifier adopted an LZ1105-16. The main parameters are shown in Table 4.

The charge amplifier used in this research can select seven gain factors which can be set by the four bits, as shown in Figure 6. In this paper, the gain factor was 1 as the sensitivity was 100 Pc/mV.

### 3.3. Data Acquisition

This experiment collected the voltage signal of the piezoelectric sensor converted by the charge amplifier. A total of 88 groups of data were collected. The original output data of the piezoelectric sensor are shown in Table 5 and Table 6. Load *F* is the original input pressure, and *U* is the output voltage of piezoelectric ceramic sensor at different temperatures. The diagram in Figure 7 is the curve of two piezoelectric sensors before compensation. Based on the output results of a 20 °C room temperature, the relative measurement error *δ* percentage of the sensors at different temperatures was calculated, according to Equation (1), as shown in Figure 8, which shows that the measurement accuracy of the sensor is greatly affected by temperature. Without any compensation, the maximum error of SP-1 was more than 35% in the temperature range of −25–80 °C, and the maximum error of SP-2 was as high as 50%. According to Equation (1):(1)$$\delta =\frac{\Delta }{L}\times 100\%$$
where:Δ is the is the absolute error;L is the true value.

## 4. Temperature Compensation Data Processing

### 4.1. Mathematical Model of Temperature Characteristics of PZT Sensor

Based on the existing experimental research, this paper established the mathematical model of temperature characteristics of a piezoelectric sensor. When the sensor size and material parameters were determined and the environment temperature was constant, the output signal had a linear relationship with the force, which can be simplified by the following equations:(2)U=KF
where:K is the sensitivity coefficient of the sensors, in V/N;F is the force applied on the sensor, in N.

K was determined by the electrical parameters of the piezoelectric materials and load ratio of the piezoelectric ceramic sheets, which were affected by temperature and difficult to calculate using a mechanical formula. If considering the temperature effect, the output voltage of the PZT sensor was not linear with the force.

It was assumed that the sensor sensitivity changed linearly with temperature from −25 °C to 85 °C.
(3)$$K=k_{T}T+k_{0}$$
where:$k_{T}$ is the PZT sensor sensitivity temperature coefficient;T is the temperature;$k_{0}$ is the sensitivity of PZT sensor at 0 °C.

Therefore, the temperature characteristic mathematical model of piezoelectric sensor is:(4)$$U=\left(k_{T}T+k_{0}\right)F$$

### 4.2. BP Neural Network of Temperature Compensation

The temperature compensation principle using a BP neural network is shown as follows: the voltage collected after temperature control of piezoelectric ceramic sensor was taken as the output value. At this time, the voltage was affected by temperature, and the voltage and temperature were taken as the input of the BP neural network. After the training of the BP neural network, the corrected pressure value was obtained. At this time, the three-layer BP neural network [29,38] was used as the training model, including the input layer, hidden layer and output layer, as shown in Figure 9.

The training of GA-BP neural networks in this study mainly included the following steps:(1)Preparation of the dataset: In this paper, 88 groups of data, obtained from SP-1 in the above experiment, were used for model construction. All of the samples were randomly divided into two sets, of which 77 groups of data were used as the training set and 11 groups of data were used as the verification set. All the data were normalized to unify their values between [–1,1];(2)Number of implied layers: The number of hidden layer nodes and the number of iterations in the training process had a great impact on the accuracy and efficiency of network training. Empirical formulas were used to calculate the hidden layer nodes;
(5)  hiddennum=sqrt(m+n)+a
where:m is the input of the layer number;n is the number of the output layer;a is generally taken as an integer between 1 and 10.
(3)Determination of the training parameters: The essence of the training process is to iteratively reduce the error between the predicted value and the target (actual) value. The learning rate may affect the training accuracy and training speed of the network. With the decrease in learning rate, the training accuracy is improved at the cost of increasing training time. However, too high a learning rate will lead to network instability and lead to failure to converge. The learning rate is usually set between 0.001 and 1. This time, the learning rate was set to 0.01. The target training error was set to 0.00001 and the number of iterations was set to 1000;(4)Genetic algorithm solution: A population size of 30 was established by a genetic algorithm; the number of iterations was 50, the mutation probability was 0.2 and the crossover probability was 0.8.

## 5. Temperature Compensation Data Processing

### 5.1. Compensation Results of the Polynomial Regression Algorithm

The simplified equation of the temperature characteristics of the piezoelectric ceramic sensor established in this paper is shown in Equation (3), and the actual fitting process is carried out in Equation (6).
(6)Z=(a×x+b)×y

In this paper, the experimental results of the sensor output temperature characteristics were fitted by MATLAB. The *X*-axis is the temperature variable, the *Y*-axis is the pressure variable, and the *Z*-axis is the corresponding output of the sensor. The fitting results are shown in Figure 10. The black spot is the measured value of the sensor test, and the surface is the fitting result. The fitting results of the two groups of sensors are shown in Table 7.

### 5.2. GA-BP Neural Network Compensation Results

The neural network structure after GA-BP training is shown in Figure 11. There were 11 samples in the training set. The data outputs of BP and GA-BP neural network for training samples are shown in Table 8. The error and predicted results of BP and GA-BP neural network are shown in Figure 12 and Figure 13. The mean square error (MSE) of BP neural network is 0.66604 and the root mean square error (RMSE) is 0.81611; the MSE of the optimized neural network is 0.017518; and the RMSE is 0.13236. It can be seen that the prediction accuracy of the GA-BP neural network is much higher than BP neural network on training set.

In order to calculate the accuracy and sensitivity measurement of the compensated piezoelectric ceramic sensor, all the data were taken into the trained model. Figure 14 shows that the genetic algorithm reached the optimal individual fitness at the 10th time with a value of 0.02811. Figure 15 shows the comparison of between the predicted value and the real value. Figure 16 shows the relative error of the two neural networks of all the samples. GA-BP neural network is much better than the BP neural network on the whole.

### 5.3. Comparison of Compensation Results and Error Analysis

The SP-2 data were compensated by the same algorithm, and the results of the two compensation algorithms are shown in Table 9. The evaluation index of the algorithm adopted the determination coefficient R^2^ and RMSE. R^2^ is an important statistic reflecting the goodness of fit of the model. It is the ratio of the sum of regression squares to the sum of total squares. Its value reflects the relative degree of regression contribution. R^2^ is the most commonly used index to evaluate the goodness of fit of the regression model. The greater R^2^ (close to 1), the better the fitted regression equation. RMSE is a measure of the deviation between the observed value and the real value. It is often used as a standard to measure the prediction results of machine learning models. To a certain extent, it can determine whether the predicted value has achieved the expected effect. The higher the RMSE value, the stronger the model’s ability to interpret information for data samples, and closer it is to the real situation. It can be seen from the table that the compensation effect of the GA-BP neural network is much better than that of polynomial linear fitting from the analysis of R^2^ and RMSE. It can be seen that the strong nonlinear mapping ability of the neural network can be widely used in the field of temperature compensation.

The calculated load *F’* of the two groups of sensors compensated by the GA-BP neural network is shown in Table 10 and Table 11. The compensated data are shown in Figure 17. It can be seen that the measurement results at different temperatures after compensation are basically unchanged. The relative error of the compensated data was calculated, as shown in Figure 18. The maximum measurement relative error of SP-1 decreased from 35.17% to 5.18%, and the maximum measurement relative error of SP-2 decreased from 50.00% to 5.71%.

In order to measure the performance index of the sensor after compensation, the relative errors of measurement accuracy, temperature sensitivity coefficient, and full scale were calculated, respectively. The relative errors of measurement accuracy before and after compensation are shown in Figure 8 and Figure 18, respectively.

The relative error percentage before and after compensation was averaged to obtain the relative error percentage of the sensors at different temperatures. The maximum average relative error of the two groups of sensors was reduced to within 2% after correction. As shown in Figure 19, the relative error of the piezoelectric ceramic sensor decreases after GA-BP compensation, and the average relative error of the sensor basically remains unchanged with the increase in temperature.

Then, the temperature sensitivity coefficient and the relative error were calculated. The sensitivity coefficient α_s_ is shown in Equation (7), and the relative error *ο* at full scale is shown in Equation (8):(7)$$\alpha_{s}=\frac{U_{\max}-U_{\min}}{\left(T_{2}-T_{1}\right)U_{\max}}$$
(8)$$ο=\frac{U_{\max}-U_{\min}}{U_{\max}}$$
where:$U_{\max}$ is the maximum values of different temperatures under the same calibration pressure;$U_{\min}$ is the minimum value.

Among them, $U_{max}-U_{min}$e the maximum value, *T*_2_ represents the upper value, *T*_1_ represent the lower value.

Using the above equations, the sensitivity coefficient before and after compensation and the relative error at full scale were calculated, respectively. The calculation results are shown in Table 12.

## 6. Conclusions

When the PZT sensor is used as a dynamic-weighing monitoring device, the improvement in detection accuracy and sensitivity is greatly limited due to the influence of temperature on the polarization intensity of the piezoelectric materials. In this paper, a PZT sensor suitable for a road traffic environment was designed, its temperature characteristics were studied, and the influence of the law of temperature on the performance of the sensor was analyzed according to the experimental results. After analyzing the results of the two compensation algorithms, the following conclusions can be drawn:(1)In this study, a piezoelectric sensor for a road Weigh-In-Motion (WIM) system was designed with piezoelectric ceramic as the core material. The sensor had the advantages of small volume, high reliability and easy construction;(2)In this paper, polynomial fitting and a GA-BP neural network were used to compensate for the output results. Compared with R^2^ and RMSE, the compensation effect of the GA-BP neural network was far better than that of polynomial linear fitting compensation, and the temperature compensation effect was obvious; this shows that the GA-BP temperature compensation model can better weaken the influence of temperature on sensor output;(3)In order to measure the performance index of the sensor after compensation, the temperature sensitivity drift of SP-1 was reduced from 1.0192 × 10^−2^ °C^−1^ to 1.896 × 10^−4^ °C^−1^, and the relative error at full scale was reduced from 30.5% to 1.42%, which was greatly improved. SP-2 also achieved the same effect. The measurement accuracy and sensitivity of piezoelectric ceramic sensor were improved.

At present, research on the temperature characteristics of piezoelectric sensors and temperature compensation methods for piezoelectric sensors are still in their initial stages. Compared with the commonly used temperature compensation methods, this paper used a genetic algorithm to optimize a BP neural network, in order to improve the accuracy and sensitivity of temperature compensation. When the new piezoelectric ceramic sensor with temperature compensation is used as the dynamic weighing device, its detection accuracy can be effectively improved.

## Figures and Tables

**Figure 1 sensors-22-02396-f001:**
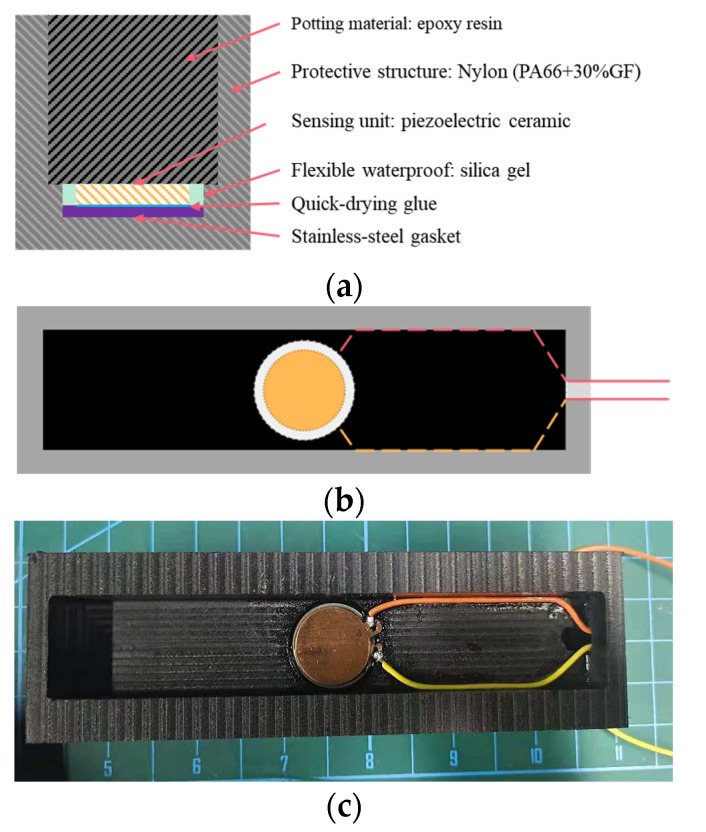
Sensor structure diagram: (**a**) internal structure; (**b**) internal layout plan; (**c**) internal real layout.

**Figure 2 sensors-22-02396-f002:**
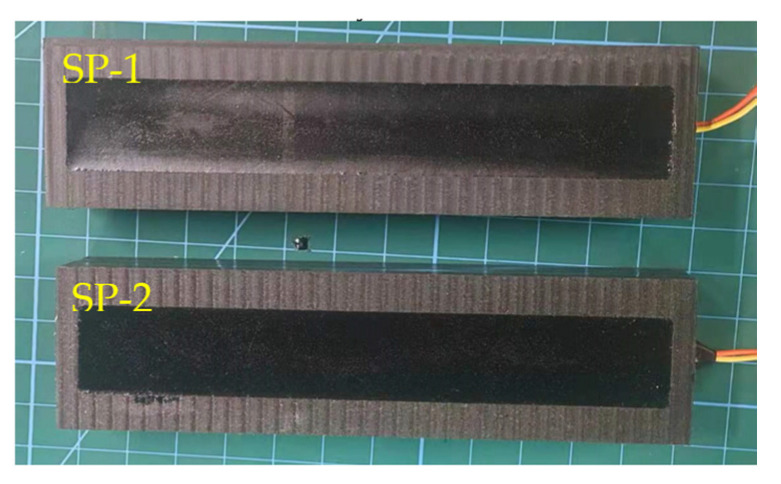
Prepared sensor samples SP-1 and SP-2.

**Figure 3 sensors-22-02396-f003:**
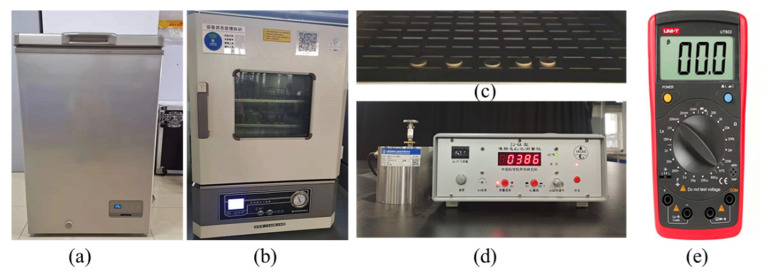
Schematic diagram of the performance measurement of piezoelectric materials: (**a**) Refrigerator; (**b**) Vacuum-drying oven; (**c**) PZT-4 Patchs; (**d**) ZJ-6A d_33_/d_31_ measuring instrument; (**e**) UT603 capacitance inductance meter.

**Figure 4 sensors-22-02396-f004:**
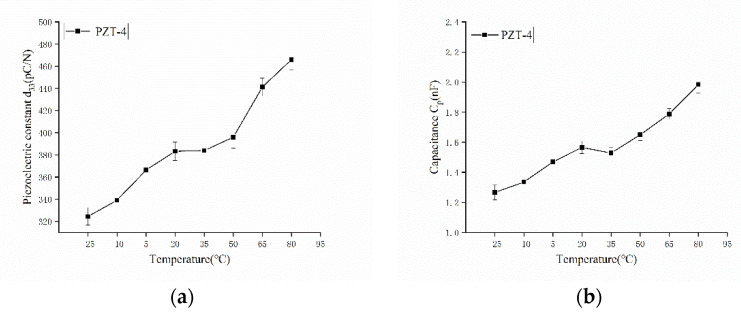
Effects of temperature on the key parameters of Piezoelectric ceramic PZT-4: (**a**) the piezoelectric coefficient d_33_; (**b**) the capacity C_p_.

**Figure 5 sensors-22-02396-f005:**
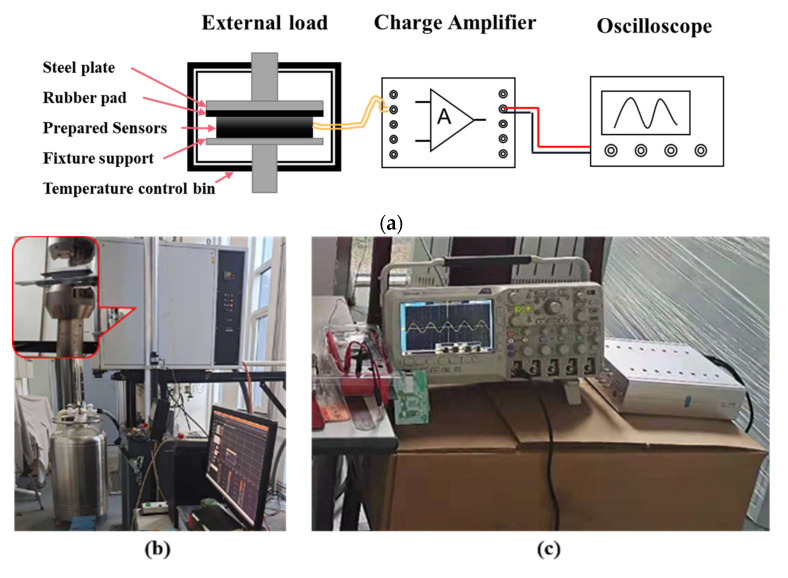
Sensor load test: (**a**) test schematic; (**b**) loading and temperature control device; (**c**) data acquisition equipment.

**Figure 6 sensors-22-02396-f006:**
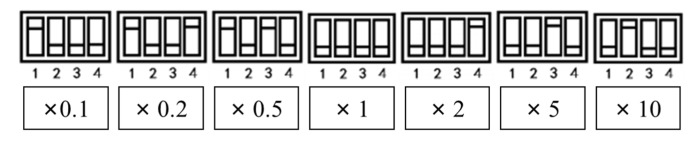
Gain adjustment of charge amplifier.

**Figure 7 sensors-22-02396-f007:**
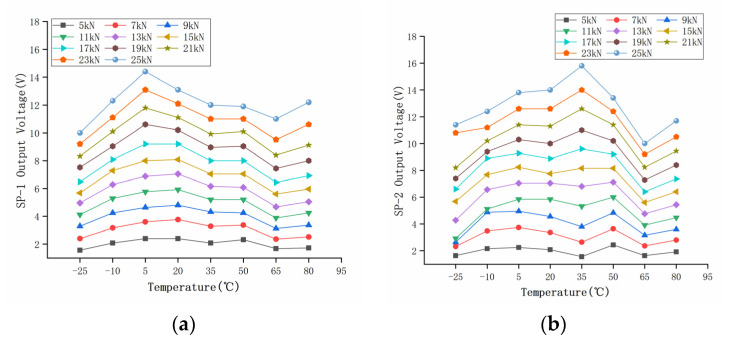
Sensor output temperature characteristics. (**a**) SP-1; (**b**) SP-2.

**Figure 8 sensors-22-02396-f008:**
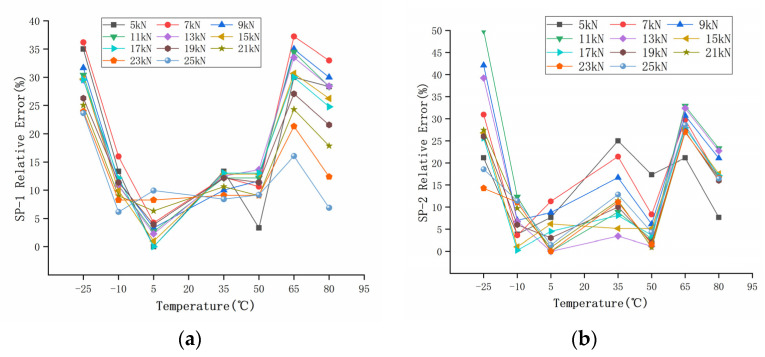
Relative error percentage of measurement at different temperatures: (**a**) SP-1; (**b**) SP-2.

**Figure 9 sensors-22-02396-f009:**
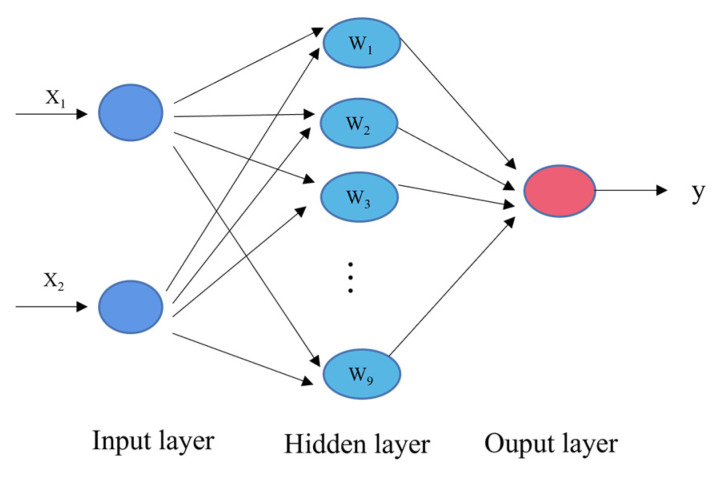
Neural network structure algorithm.

**Figure 10 sensors-22-02396-f010:**
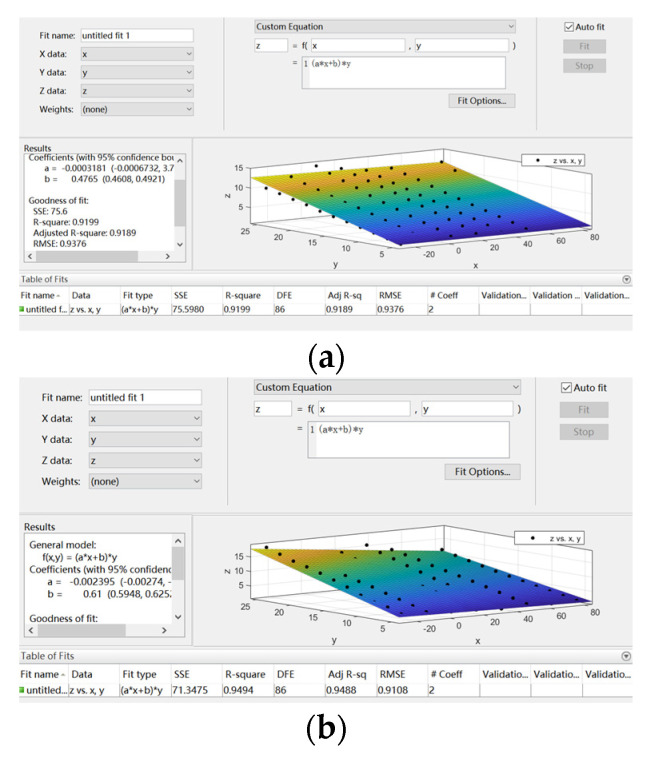
Temperature-fitting surface: (**a**) SP-1 fitting surface; (**b**) SP-2 fitting surface.

**Figure 11 sensors-22-02396-f011:**
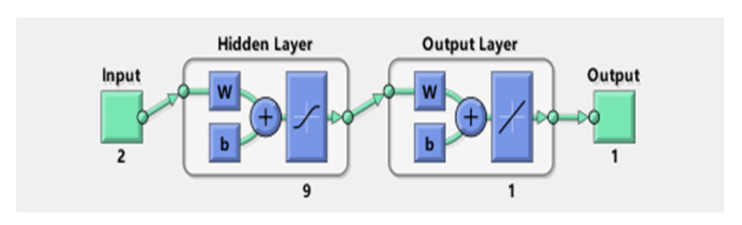
Structural diagram of the neural network.

**Figure 12 sensors-22-02396-f012:**
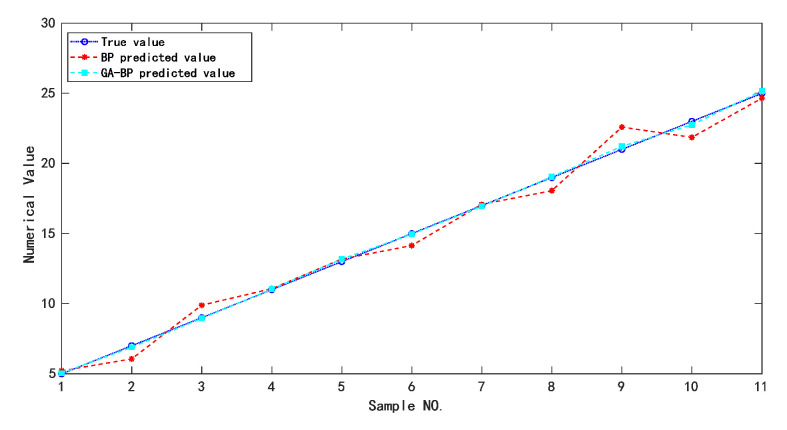
Comparison of predicted value and real value for the training samples.

**Figure 13 sensors-22-02396-f013:**
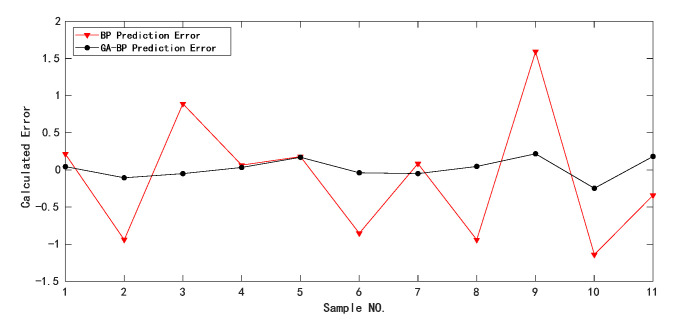
Comparison of relative errors of prediction results for the training samples.

**Figure 14 sensors-22-02396-f014:**
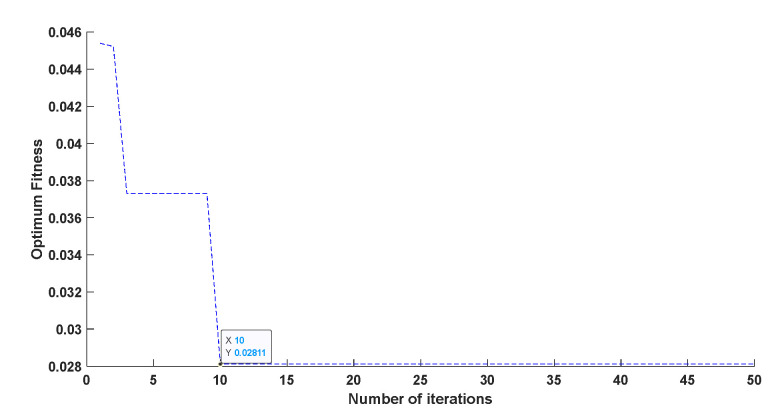
Curve of optimum fitness.

**Figure 15 sensors-22-02396-f015:**
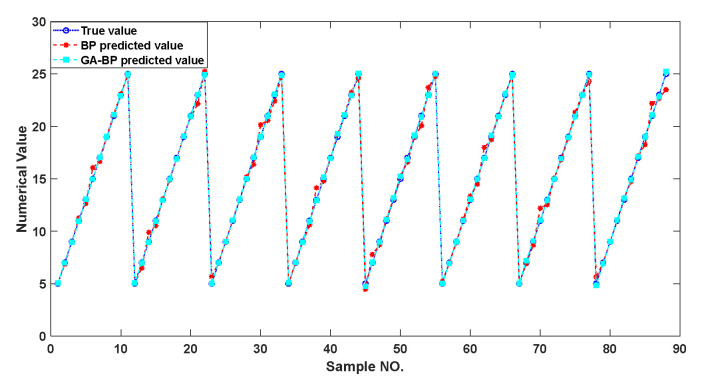
Comparison of predicted values and real values for all samples.

**Figure 16 sensors-22-02396-f016:**
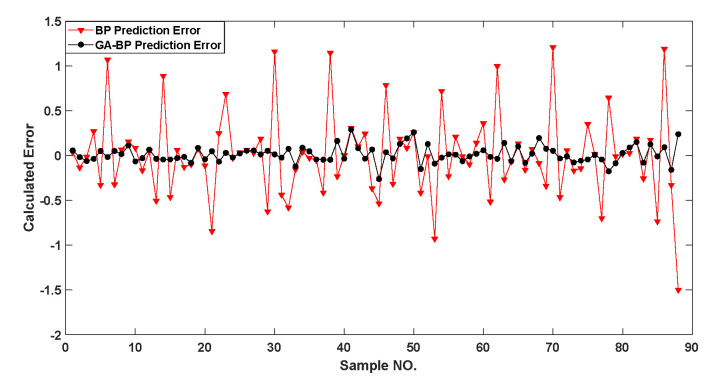
Comparison of relative errors of prediction results for all samples.

**Figure 17 sensors-22-02396-f017:**
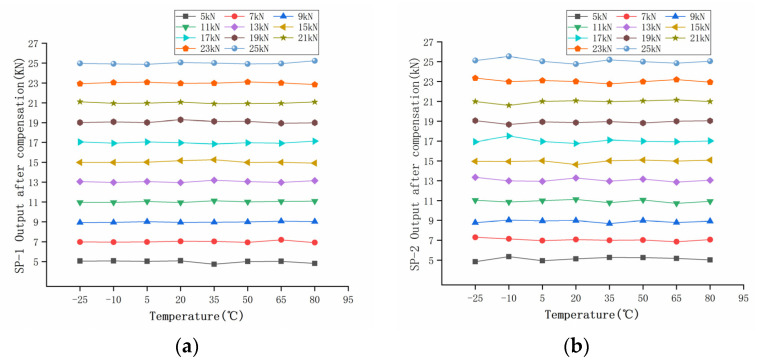
Temperature characteristics after compensation: (**a**) SP-1 output after compensation; (**b**) SP-2 output after compensation.

**Figure 18 sensors-22-02396-f018:**
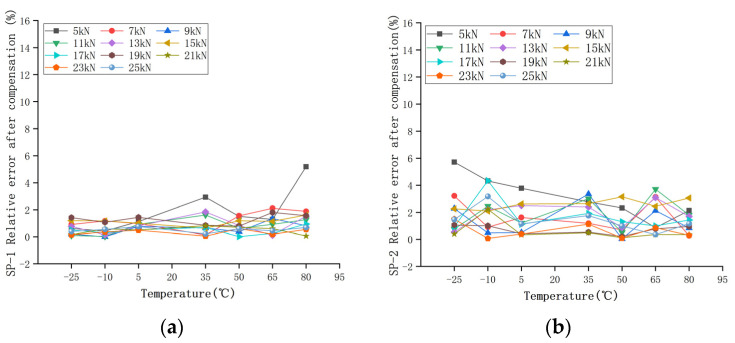
Error percentage at different temperatures after compensation: (**a**) SP-1; (**b**) SP-2.

**Figure 19 sensors-22-02396-f019:**
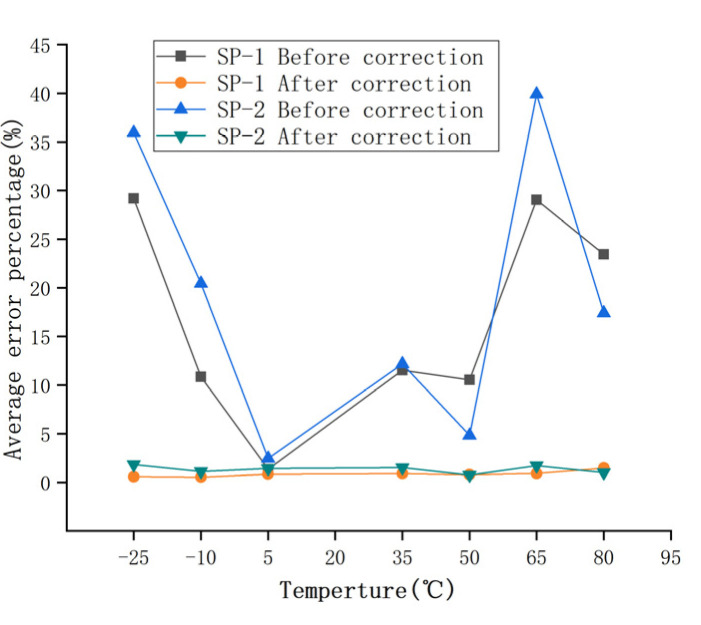
Average relative error before and after compensation.

**Table 1 sensors-22-02396-t001:** Properties of PZT-4.

Parameters	Symbols	Values	Units
Piezoelectric constant	$d_{33}$	220	pC/N
Relative dielectric constant	$\varepsilon_{33}^{T}$	1050	-
Curie temperature	$T_{c}$	310	°C
Density	ρ	7.45	10^3^ kg/m^3^
Elastic modulus	E	83.3	GPa
Electromechanical coupling coefficient	Q_m_	>0.63	-

**Table 2 sensors-22-02396-t002:** Properties of glass-fiber-reinforced nylon (PA66+30%GF).

Parameters	Test Standard	Mechanical Property
Density	ISO1183	1.48 g/cm^−3^
Tensile strength	ISO527	145 MPa
Elongation at break	ISO527	2%
Bending strength	ISO178	200 MPa
Elastic modulus	ISO604	5900 MPa
Poisson’s ratio	ISO527	0.34

**Table 3 sensors-22-02396-t003:** Epoxy resin material parameters.

Parameters *	Component A	Component B
Density	1.45–1.80 g/cm^3^	1.08–1.15 g/cm^3^
Dynamic viscosity	3000–4000 cps	100–250 cps
Hardness (Shore-D)	≥80	
Insulation strength	≥1015 Ω·cm	
Volume resistivity	≥15 V/mm	
Shear strength	≥10 MPa	
Dielectric constant	3.0 ± 0.1	
Operating temperature	−30–120 °C	
Curing shrinkage	≤0.5%	

* These parameters were measured at 25 °C.

**Table 4 sensors-22-02396-t004:** Main parameters of charge amplifier.

Parameters	Units	Value
Measuring charge ranges	pC	±10^7^
Output voltage ranges	mV	±10,000
Gain factor		0.1, 0.2, 0.5, 1, 2, 5, 10
Sensitivity *	pC/mV	100
Measurement uncertainty	%	0.5
Frequency range	Hz	0.1–200,000

* This parameter is measured when gain factor is 1.

**Table 5 sensors-22-02396-t005:** SP-1 raw output data.

Load (kN)	U(V) T = −25 °C	U(V) T = −10 °C	U(V) T = 5 °C	U(V) T = 20 °C	U(V) T = 35 °C	U(V) T = 50 °C	U(V) T = 65 °C	U(V) T = 80 °C
5	1.56	2.08	2.40	2.40	2.08	2.32	1.68	1.72
7	2.40	3.16	3.60	3.76	3.28	3.36	2.36	2.52
9	3.28	4.24	4.64	4.80	4.32	4.24	3.12	3.36
11	4.12	5.28	5.76	5.92	5.20	5.20	3.88	4.24
13	4.96	6.28	6.88	7.04	6.16	6.08	4.68	5.04
15	5.68	7.28	8.00	8.08	7.04	7.04	5.60	5.96
17	6.48	8.08	9.20	9.20	8.00	8.00	6.44	6.92
19	7.52	9.04	10.6	10.20	8.96	9.04	7.44	8.00
21	8.32	10.10	11.8	11.10	9.92	10.10	8.40	9.12
23	9.20	11.10	13.1	12.10	11.00	11.00	9.52	10.60
25	10.00	12.30	14.4	13.10	12.00	11.90	11.00	12.20

**Table 6 sensors-22-02396-t006:** SP-2 raw output data.

Load (kN)	U(V) T = −25 °C	U(V) T = −10 °C	U(V) T = 5 °C	U(V) T = 20 °C	U(V) T = 35 °C	U(V) T = 50 °C	U(V) T = 65 °C	U(V) T = 80 °C
5	1.64	2.16	2.24	2.08	1.56	2.44	1.64	1.92
7	2.32	3.48	3.74	3.36	2.64	3.64	2.36	2.80
9	2.64	4.88	4.96	4.56	3.80	4.84	3.16	3.60
11	2.92	5.12	5.84	5.84	5.32	6.00	3.92	4.48
13	4.28	6.56	7.04	7.04	6.80	7.12	4.76	5.44
15	5.68	7.68	8.24	7.76	8.16	8.16	5.60	6.40
17	6.60	8.90	9.28	8.88	9.60	9.20	6.40	7.36
19	7.40	9.40	10.30	10.00	11.00	10.20	7.28	8.40
21	8.20	10.20	11.40	11.30	12.60	11.40	8.24	9.44
23	10.80	11.20	12.60	12.60	14.00	12.40	9.20	10.50
25	11.40	12.40	13.80	14.00	15.8	13.40	10.00	11.70

**Table 7 sensors-22-02396-t007:** Polynomial regression algorithm fitting the results.

Number	Parameter Values	Fitting Equation	R^2^	RMSE
SP-1	a = −3.18 × 10^−^^4^ b = 0.4765	U = (−3.18 × 10^−^ ^4^T + 0.4765) × F	0.9199	0.9376
SP-2	a = −2.93 × 10^−^^3^ b = 0.61	U = (−2.93 × 10^−^^3^T + 0.61) × F	0.9494	0.9108

**Table 8 sensors-22-02396-t008:** **Output** Data of SP-1 before and after neural network optimization.

Number	Measured Value	BP Predicted Value	GA-BP Value	BP Error	GA-BP Error
1	5.0000	5.2144	5.0446	0.2144	0.0446
2	7.0000	6.0610	6.8939	−0.9390	−0.1061
3	9.0000	9.8911	8.9507	0.8911	−0.0493
4	11.0000	11.0654	11.0332	0.0654	0.0332
5	13.0000	13.1809	13.1708	0.1809	0.1708
6	15.0000	14.1470	14.9619	−0.8530	−0.0381
7	17.0000	17.0847	16.9517	0.0847	−0.0483
8	19.0000	18.0599	19.0481	−0.9401	0.0481
9	21.0000	22.5921	21.2169	1.5921	0.2169
10	23.0000	21.8608	22.7545	−1.1392	−0.2455
11	25.0000	24.6584	25.1824	−0.3416	0.1824

**Table 9 sensors-22-02396-t009:** Comparison of the results of the compensation algorithm.

Number	Polynomial Linear Fitting Compensation		GA-BP Neural Network Compensation	
	R^2^	RMSE	R^2^	RMSE
SP-1	0.9199	0.9376	0.9993	0.0936
SP-2	0.9494	0.9108	0.9988	0.2251

**Table 10 sensors-22-02396-t010:** SP-1 GA-BP neural network output after training.

Input Load (kN)	*F’* (kN) T = −25 °C	*F’* (kN) T = −10 °C	*F’* (kN) T = 5 °C	*F’* (kN) T = 20 °C	*F’* (kN) T = 35 °C	*F’* (kN) T = 50 °C	*F’* (kN) T = 65 °C	*F’* (kN) T = 80 °C
5	5.06	5.07	5.03	5.09	4.94	5.01	5.02	4.82
7	6.98	6.96	6.98	7.05	7.04	6.94	7.19	6.91
9	8.94	8.96	9.02	8.96	8.97	8.99	9.08	9.03
11	10.96	10.96	11.05	10.95	11.13	11.02	11.05	11.09
13	13.05	12.97	13.06	12.95	13.19	13.06	12.97	13.15
15	14.98	14.99	15.01	15.16	15.26	14.98	14.99	14.92
17	17.05	16.92	17.05	16.97	16.85	16.96	16.92	17.12
19	19.02	19.08	19.01	19.29	19.13	19.14	18.94	18.99
21	21.11	20.96	20.98	21.08	20.91	20.94	20.95	21.09
23	22.93	23.05	23.07	22.96	22.98	23.10	23.01	22.84
25	24.97	24.93	24.88	25.07	25.01	24.92	24.96	25.24

**Table 11 sensors-22-02396-t011:** SP-2 GA-BP neural network output after training.

Input Load (kN)	*F’* (kN) T = −25 °C	*F’* (kN) T = −10 °C	*F’* (kN) T = 5 °C	*F’* (kN) T = 20 °C	*F’* (kN) T = 35 °C	*F’* (kN) T = 50 °C	*F’* (kN) T = 65 °C	*F’* (kN) T = 80 °C
5	4.84	5.36	4.94	5.14	5.28	5.26	5.18	5.03
7	7.30	7.14	6.96	7.07	6.99	7.02	6.85	7.06
9	8.79	9.04	8.95	8.99	8.69	8.99	8.80	8.92
11	11.05	10.86	11.00	11.13	10.80	11.07	10.72	10.93
13	13.35	12.98	12.94	13.27	12.96	13.16	12.86	13.05
15	14.96	14.94	15.01	14.63	15.01	15.09	14.98	15.08
17	16.93	17.50	16.96	16.77	17.10	16.99	16.94	17.02
19	19.06	18.67	18.94	18.86	18.96	18.82	19.00	19.04
21	20.99	20.60	21.00	21.07	20.97	21.05	21.15	20.99
23	23.34	22.98	23.09	23.00	22.74	22.98	23.19	22.93
25	25.12	25.54	25.04	24.76	25.19	25.00	24.85	25.05

**Table 12 sensors-22-02396-t012:** The sensitivity coefficient and the relative error calculation results at the full range.

SP-1	Before compensation	$\alpha_{s}=1.0192\times 10^{-2}$ °C^−1^	o=30.5%
	After compensation	$\alpha_{s}=1.896\times 10^{-4}$ °C^−1^	o=1.42%
SP-2	Before compensation	$\alpha_{s}=0.635\times 10^{-2}$ °C^−1^	o=28.57%
	After compensation	$\alpha_{s}=4.078\times 10^{-4}$ °C^−1^	o=3.67%

## Data Availability

All datasets used in this study are discussed in Section 3 and Section 5. They are publicly available and cited in the list of references.

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
