# Peer review of "Investigation of the Temperature Compensation of Piezoelectric Weigh-In-Motion Sensors Using a Machine Learning Approach"

_sensors, 2022, doi:10.3390/s22062396_

Round 1

Reviewer 1 Report

The paper describes an interesting and important problem
Remarks:
1. On line 101, the font size seems different than it should be
2. Tables 10 and 11 should change the headings F (kN) so that the unit with parentheses is on one line. Can make the font smaller.
3. The patterns that are included in the articles are not marked whether they come from lliterature or are "invented" by the authors
4. In figure 1 there is a piezoelectric diagram without marking where is the glue, which is in the description for this figure

Author Response

Thank very much for your comments. The author has carefully revised the article and responded to your questions according to your comments.Please see attachment for details.

Reviewer 2 Report

Comments and Suggestions for the Authors

This work, titled “Investigation on the temperature compensation of piezoelectric Weigh-In-Motion Sensors using a machine learning approach”, reports on the design and characterization of a WIM system to automatically detect vehicles load information by means of load cells embedded on the driving road. Moreover in the paper, the authors discuss about an interesting method to compansate the dependence on the temperature of the ceramic piezoelectric materials on which the sensors themselves are based. In order to compensate the influence of temperature on piezoelectric sensor and improve the detection accuracy and sensitivity the compensation is based on a mathematical model of temperature characteristics of piezoelectric ceramic sensor and Genetic Algorithm Back Propagation (GA-BP) neural network model. The experimental results show that the GA-BP neural network temperature compensation model can improve the measurement accuracy and sensitivity of piezoelectric ceramic sensor.

The reviewer found this work interesting and useful. However, major revisions are needed before being further considered for publication in Sensors. Please, address the following comments:

  1. At line 103 of page 3, the authors talk about “environmental adaptability” of the piezoelectric ceramic sensor. Could the authors better explain this concept?
  2. In table 1 at page 3, the authors reported the properties of the materials used to fabricate and assemble the final sensor, providing details especially on the piezoelectric material. Nevertheless, the reviewer think useful to include in the table information about the Elastic Modulus of both the glass-fiber and the epoxy resin too. Could the authors address this comment?
  3. At lines 111 – 117 of page 3, the authors describes the sensor packaging. Nevertheless, there are different annotation in detailing the patch size. Please, could the authors use the same annotation, e.g. 20 mm x 2mm (instead 20x 2mm) as 25mm x 2mm? Moreover, what does the label j describe in the text?
  4. At lines 118 – 122 of page 4 and lines 148 – 152 of page 5, the authors describe the manufacturing process of the sensor and the preliminary tests under different temperature. For the sake of comprehension and text clarity, the reviewer suggests to adjust the English form of the test, using the same tense (present or past).
  5. At page 5, the authors describe the temperature test to analyse the changes of the piezoelectric coefficient of the piezoelectric ceramics. Which system was used to change and control the temperature from -25° up to 80°? The results of this test are reported in figure 4. Did the authors calculate also the standard deviation of the measured piezoelectric constant and capacitance at each temperature? The reviewer suggests to include this value to improve the quality of the graph.

For the sake of clarity, please add some literature references to the loading force range indicated at line 178, if this range is not a result of calculation. Please also correct the tense in the sentence “…which is covers the range”.

  1. In order to perform voltage measurements, the authors used a charge amplifier to convert the measured charge and amplify the final signal. Please, could the authors provide more details about the charge amplifier? Which is the charge-to-voltage conversion factor?
  2. In tables 5 and 6, the raw outputs coming from the sensor SP1 and SP2 are listed. Nevertheless, the output voltages seem to be affected by a variability at the same temperature. Which is the difference between the two sensor SP1 and SP2. Can this variability influence the final results?
  3. At page 8, the authors show the graph of the relative error percentage. The reviewer think it is useful to include in the text an equation of the calculated relative error percentage.
  4. In table 10 and table 11, what does the label F’ refer to? Please, address this comment.
  5. At line 368 – 391 of page 15, the authors show a comparison between the relative error percentage before and after compensation. There is still here a difference of the relative calculated error of about 2.5 times between the sensor SP1 and SP2, even after compensation. Could the authors explain the reasons of this difference?

Moreover, please address the following minor revisions:

  1. Line 45, correct reference 1 annotation;
  2. Line 48, replace “rang-e” with “range”;
  3. Line 76, replace comma “,” with ending point “.”;
  4. Line 291, adjust Figure 9 font;
  5. Line 351, adjust Figure 9 font;
  6. Line 372, replace comma “,” with ending point “.”

Author Response

Thank very much for your comments. The author has carefully revised the article and responded to your questions according to your comments. Please see attachment for details.

Reviewer 3 Report

This paper is to describe a temperature compensation method of piezoelectric WIM sensors with a genetic algorithm back propagation. This is well organized with good explanation, and could be recommended to publish in the Sensors with minor revisions.

  1. The quasi-static 154 piezoelectric coefficient system (ZJ-6A) has temperature control function? And the data of Figure 4 should have error bar?
  2. The charge amplifier of Figure 5 should be explained in detail!
  3. The piezoelectric ceramic has the hysteretic loop between input F and output V. The equation (1) is approximately good, but is not perfect for high precision measurement.
  4. The format should be revised.

Author Response

(The authors gave the same response as above.)

Round 2

Reviewer 2 Report

The authors comprehensively replied to the reviewer comments. For this reason, the reviewer think the paper can be accepted in present form and  considered for publication in Sensors.